# Contribution of Pro-Inflammatory Molecules Induced by Respiratory Virus Infections to Neurological Disorders

**DOI:** 10.3390/ph14040340

**Published:** 2021-04-08

**Authors:** Karen Bohmwald, Catalina A. Andrade, Alexis M. Kalergis

**Affiliations:** 1Departamento de Genética Molecular y Microbiología, Millennium Institute on Immunology and Immunotherapy, Facultad de Ciencias Biológicas, Pontificia Universidad Católica de Chile, Santiago 8331010, Chile; kbohmwald@uc.cl (K.B.); cnandrade@uc.cl (C.A.A.); 2Departamento de Endocrinología, Facultad de Medicina, Pontificia Universidad Católica de Chile, Santiago 8331010, Chile

**Keywords:** pro-inflammatory molecules, brain, neuropsychiatric disorders, viral infection

## Abstract

Neurobehavioral alterations and cognitive impairment are common phenomena that represent neuropsychiatric disorders and can be triggered by an exacerbated immune response against pathogens, brain injury, or autoimmune diseases. Pro-inflammatory molecules, such as cytokines and chemokines, are produced in the brain by resident cells, mainly by microglia and astrocytes. Brain infiltrating immune cells constitutes another source of these molecules, contributing to an impaired neurological synapse function, affecting typical neurobehavioral and cognitive performance. Currently, there is increasing evidence supporting the notion that behavioral alterations and cognitive impairment can be associated with respiratory viral infections, such as human respiratory syncytial virus, influenza, and SARS-COV-2, which are responsible for endemic, epidemic, or pandemic outbreak mainly in the winter season. This article will review the brain′s pro-inflammatory response due to infection by three highly contagious respiratory viruses that are the leading cause of acute respiratory illness, morbidity, and mobility in infants, immunocompromised and elderly population. How these respiratory viral pathogens induce increased secretion of pro-inflammatory molecules and their relationship with the alterations at a behavioral and cognitive level will be discussed.

## 1. Introduction

Pro-inflammatory molecules, such as cytokines (e.g., tumor necrosis alpha (TNF-α), interleukin (IL)-1β, IL-6, granulocyte-macrophage colony-stimulating factor (GM-CSF)) [1,2] and chemokines (e.g., CCL2, CCL3, CCL4, CCL5, CCL11, CXCL-8, and CXCL10) are small signaling molecules that are mainly secreted by immune cells, as well as by brain cells, including microglia, astrocytes, and neurons [3]. These molecules play important roles both in the immune system and for the central nervous system′s normal function (CNS) [4,5]. For example, cytokines participate in the cognitive process, and it has been shown that both the overexpression and the down-regulation of these cytokines alter synaptic plasticity [6,7,8]. According to this concept, the central cytokines involved in the memory and learning processes are IL-1β, IL-6, and TNF-α [7,9,10]. Reports indicate that IL-1β is expressed in the CNS at low levels under certain physiologic conditions and is required for the consolidation of the memory and learning processes [10]. Also, it has been described that an increase or the absence of IL-1β negatively affects the cognition process [6,11].

On the other hand, IL-6 is a pro-inflammatory cytokine found in the CNS under certain physiological conditions and secreted mainly by glial cells [9,12]. It is known that both IL-1β and IL-6 levels are below the limits of detection for protein levels under certain physiological conditions [12]. Moreover, it was shown that IL-6 expression is increased during the LTP process, and the consolidation of this LTP is affected by the inhibition of IL-6 using a neutralizing antibody [9]. Meanwhile, TNF-α is also a pro-inflammatory cytokine that exerts physiological, neuroprotective, and neurodegenerative effects in the CNS [6]. Accordingly, it has been previously reported that TNF-α plays a role in the induction of LTP and LTD because either the absence or overexpression of these cytokines is detrimental to the learning and memory processes [6,13].

Additionally, IFN-γ has been associated with enhanced neurogenesis in adult mice’s’ dentate gyrus and improved spatial learning and memory [6]. Besides, chemokines such as CCL2 and CX3CL1 also contribute to synaptic plasticity in the CNS [14,15,16,17]. CCL2 is constitutively expressed in several zones of the CNS, including the hippocampus, and can modulate different neuron populations′ electrical activity (for instance, hippocampal neurons) [18,19]. Furthermore, CX3CL1 is a chemokine that is constitutively expressed in the CNS, with exceptionally high levels by hippocampal neurons, and the receptor this chemokine is expressed by microglia [17]. It has been shown that CX3CL1 can modulate the glutamatergic synapsis and exert a neuroprotective effect [17,20].

Neurobehavioral alterations and cognitive impairment are part of neuropsychiatric disorders, such as autism spectrum disorder (ASD) [21], schizophrenia (SCZ) [22], mood alterations [23], learning and memory impairment, among others. The incidence of these pathologies is about 10 to 20% worldwide [24]. The etiology of these ailments is multifactorial, where not only genetic factors are relevant, but also the environment can contribute significantly to the triggering of such neuropsychiatric disorders [25]. According to this, several studies have associated the neuroinflammation caused by pro-inflammatory molecules with these neuropsychiatric pathologies [26,27,28].

Schizophrenia (SCZ) is a neurodevelopmental and chronic psychiatric disorder with multifactorial causes, characterized by alterations at the cognitive, behavioral, and emotional levels [29]. Studies have described that reducing the gray matter takes place, mainly in the hippocampus, prefrontal, and temporal cortex [30,31,32]. SCZ has four stages; risk stage, the prodrome of SCZ, the onset of psychosis, and chronic disability stage [33]. A pro-inflammatory component has been shown for this mental disease, with an abnormal serum pattern of cytokines characterized by high levels of IL-2, IL-6, and CXCL8 (also known as IL-8) [34,35,36]. It has been suggested that exposure to increased levels of pro-inflammatory molecules during pregnancy can alter neurodevelopment [37]. Elevated levels of CXCL8 during the second or third trimester of pregnancy can have an impact on the development of the fetus brain, which might lead to an alteration of the synaptic density, neuronal migration, and/or survival of these cells in the cortex. These neurological alterations can contribute to explaining the memory impairment observed in schizophrenia [37].

Meanwhile, the altered secretion of IL-6 and TNF-α during the last trimester of pregnancy has been related to the inhibition of proper synaptic formation and glial cell development [37]. The SCZ is manifested in puberty or early adulthood, where serum IL-1β, IL-6, and TNF-α increase in patients with the first episode of psychosis (FEP) with or without drug treatment [38]. CXCL8 and IFN-γ also have been found elevated in sera from patients with FEP with drug treatment and stable chronic patients [38]. However, these data are controversial due to depending on the experimental design and sample collection of the study.

Another neuropsychiatric illness is autism spectrum disorder (ASD), a behavioral alteration mainly in communication and social interaction, besides the stereotyped behavior with restricted interest [39,40]. The etiology of this disease is heterogeneous, but it is thought that maternal pro-inflammatory molecules′ contribution may contribute to the neurodevelopment of the fetus brain [41,42,43]. Pro-inflammatory cytokines, such as GM-CSF, IL1-α, IL-6, and IFN-γ increase in the serum during the mid-gestational maternal, correlates with a higher risk of ASD and intellectual disability [41]. Furthermore, cytokines that include IL-1β, IL-5, and IL-13, among others, have also been implicated in the severity of ASD after birth [44]. During the first ten years of age, studies have been shown that ASD children have increased level of pro-inflammatory molecules anti-inflammatory molecules such as IL-4 and IL-13 [45,46,47].

In the past few years, the hypothesis that viral infections can be, in part, responsible for the development of neurological illnesses has been explored [25,48]. Consistently, some viruses invade and cause damage to the central nervous system (CNS) directly by infecting the resident cells or by a local inflammation induced for the host immune response [25,49]. Most of the viruses infecting the CNS can spread from the infection site to the peripheral nervous system and travel via axon fibers to the CNS areas, through olfactory nerves and the epithelium [25,50,51]. Viruses can also enter the CNS via a ‘Trojan horse’ mechanism, in which infected leukocytes can carry pathogens from the blood across the blood–brain barrier (BBB) [50,51]. The BBB is a multicellular structure that is part of the neurovascular unit (NVU) and is composed of endothelial cells, pericytes, astrocytes, neurons, and the basement membrane located near the cerebral vasculature [52]. The BBB function is maintenance homeostasis by controlling the passage of macromolecules, cells, and foreign substances, mediating the communication between the periphery and the CNS [52,53]. The endothelial cells are attached to each other through the tight junction that allows forming a monolayer that acts as the first line of defense against circulating molecules and foreign cells [53]. The endothelial cells are attached to each other through the tight junction that allows forming a monolayer that acts as the first line of defense against circulating molecules and foreign cells [53].

On the other hand, astrocytes are the most abundant cell type of the CNS, which has multiple functions, such as pH maintenance, homeostasis of ion concentrations, participates in blood flow regulation, and participates in neurotransmission, the local immune response, among others [53,54]. The pericytes are located in the abluminal surface of the endothelial cells and the vascular basement membrane. These cells are connected with the endothelial cells through gap junctions and are necessary to develop tight junctions even in the uterus′ BBB function [53]. Furthermore, pericytes participate in the BBB integrity, angiogenesis, and microvasculature stability, among others [55].

As was mentioned above, the BBB controls the passage of cells into the CNS. The peripheral immune surveillance is performed by specialized innate immune cells such as macrophages located in the meninges, perivascular spaces, and choroid plexus [56]. Moreover, it is known that monocytes, dendritic cells, and T lymphocytes are present in the cerebrospinal fluid (CSF) of healthy individuals, where the most abundant subset found are activated central memory T cells which monitoring the subarachnoid space [57,58,59,60]. Additionally, these T cells maintain the ability to initiate a local immune response, or in the absence of an injury, these cells can return to a secondary lymphoid organ [59].

The CNS local immune response is commanded by microglia that are the only immune cells found in the brain parenchyma [58,61]. Microglia patrol the healthy CNS and when they found strange signals, increased HLA-DR expression in humans (MHC-II in mouse) due to the antigen presentation capacity [58]. Moreover, microglia can be activated and, depending on the stimuli, polarize to an M1 or M2 phenotype, which secretes pro-inflammatory and anti-inflammatory molecules, respectively [62]. On the other hand, astrocytes also contribute to the CNS immune response, which can detect dangerous signals and respond to secreting pro-inflammatory molecules at the same as microglia. Astrocytes have a cytoskeletal protein named glial fibrillary acid protein (GFAP) that not only allows identifying these cells but also its expression increased where astrocytes are activated [63]. It was established that both glial cells respond quickly to a CNS insult and that the interaction between astrocytes and microglia is crucial for the microglial immune response [64].

It is known that some viruses, such as influenza, hRSV, coronaviruses, among others, can complete their viral cycle in CNS cells and further cause alterations in normal function [25,65,66,67]. However, viral infections can also cause CNS inflammation without brain invasion and lead to alterations in this tissue′s function that could cause neurological damage.

## 2. Neuroinflammation Induced by Respiratory Viruses

As described above, neuroinflammation can contribute to several behavioral and cognitive disorders [26,27]. Among the most relevant respiratory viruses leading to inflammation in the CNS are the influenza virus, the human respiratory syncytial virus (hRSV), and SARS-CoV2 [68,69,70]. The influenza virus′s relevance is its capability to infect people of a wide range of ages during the winter season and can cause neurological alteration from pregnancy [71,72]. As for the case of hRSV, studies have shown that it is a highly contagious virus that affects mainly infants, which can affect neurological development [73]. Although SARS-CoV2 is an emerging virus that mainly infects young adults and the elderly, it has been shown to affect the CNS [74,75,76]. In this section, we will discuss how the infection by these viruses can promote neuroinflammation and the consequences for the host.

### 2.1. Influenza Virus

Influenza virus is the most important viral agents causing acute respiratory tract infections in a wide range of ages, which affects between 2% to 10% of the global population [77]. Every year influenza viruses causes around 500,000 death worldwide [77]. Influenza viruses belong to the Orthomyxoviridae family, which includes at least four members; Influenza A (IAV), B (IBV), C (ICV), and D (IDV) [78,79,80]. Only IAV and IBV are responsible for seasonal epidemics annually [79]. The Influenza virus′s viral structure is composed of a segmented single-strand RNA with negative-sense, which has an envelope [79]. The genomes of IAV and IBV consist of eight segments, while ICV has seven segments [80]. The classification of IAV subtypes is based on two structural proteins, hemagglutinin (HA) and neuraminidase (NA), which can be found in eighteen different HA subtypes (H1-H18) and eleven NA subtypes (N1–N11) [67,79]. The H1, H2 and H3, subtypes are relevant for human health because they can be transmitted between individuals and cause neurological symptoms [81,82].

An influenza virus infection can cause severe manifestations, such as pneumonia and acute respiratory distress syndrome [83,84]. The immune response against influenza virus is initiated by the airways epithelial cells, which secrete pro-inflammatory molecules such as IFNα/β, IL-6, TNF-α CCL2, CCL3, CCL5, and CXCL8, allowing the recruitment of natural killer (NK) cells, monocytes, macrophages, dendritic cells (DCs), and neutrophils [85,86,87,88,89]. Upon influenza infection, activated CD4+ T cells polarized to both Th1 and Th2 phenotypes, but Th1 cells are associated with survival after the infection [88]. Also, CD8+ cells are responsible for the viral clearance and bring protection to the host during Influenza infection [87]. Furthermore, the Th17 cells suppress the regulatory T cells (Tregs) function through the secretion of IL-6; meanwhile, Tregs control both the CD4+ and CD8+ T cell responses after infection [88].

The clinical neurological symptoms associated with the influenza virus respiratory infection that has been described consist of encephalitis [90,91,92,93], myelitis [84,94], meningitis [95,96,97], seizures [93,96] and Guillain-Barré syndrome [97,98]. Moreover, influenza virus infection has been related to chronic manifestations, such as SCZ [99,100], ASD [101], and mood disorders. All the evidence about the influenza virus and neurological pathologies suggests direct or indirect effects on the occurrence of neuroinflammation. Despite the neurological symptoms shown by several patients in different studies, the detection of influenza viruses in CSF has been only infrequently reported [102,103,104,105,106]. Furthermore, increased levels of pro-inflammatory molecules, such as IL-1β, IL-6, CXCL8, CCL2, CXCL10, CXCL9, and TNF-α, have been detected in the CSF of patients showing neurological alterations due to influenza virus infection, such as acute encephalitis and encephalopathy [103,107,108]. Further, increased serum levels for IL-6, IL-10, TNF-α, IFN-γ [107,108] have been shown for these patients (Figure 1).

Neuroinflammation has been described during the acute phase of the influenza virus infection [91]. As was mentioned above, besides the neurological symptoms observed in patients, not always the viral RNA can be detected. For this reason, studies performed in mice show that both neurotropic and non-neurotropic strains of the influenza virus can cause CNS alterations [111,112,113,114,115]. Mice intranasally challenged with H5N1 (Vietnam/1203/04) showed viral RNA in the CNS at three days post-infection, demonstrating the neurotropic potential of this strain [115,116]. First, the H5N1 (Vietnam/1203/04) was related to developing a parkinsonian phenotype in mice that is a neurodegenerative pathology characterized by protein aggregation, a phenomenon observed in Influenza virus-infected mice [115]. Furthermore, Influenza virus not only can reach the brain but also can infect resident cells, such as neurons and microglia. Besides, the influenza virus infection promotes microglial apoptosis [116]. Moreover, the profile of pro-inflammatory cytokines, chemokines, and growth factors was evaluated in four brain zones (brainstem, substantia nigra, striatum, and cortex) and different points post-infection (p.i.). In the brainstem, it was observed an increase of pro-inflammatory cytokines, such as IL-1β, IL-12(p70), IL-13, and TNF-α; chemokines, such as CCL2, CCL3, CCL4, CCL11, CXCL10, and CXCL1; growth factors, such as G-CSF, GM-CSF, M-CSF, and anti-inflammatory IL-10 at ten days post-infection (Table 1) [116]. Interestingly, increased levels of IL-1β, IL-2, and VEGF were found 60–90 days p.i. in the substantia nigra, elevated levels of IL-1β, IL-2, IL-6, CCL2, G-CSF, and M-CSF were found during the acute phase of influenza virus infection, whereas CCL4 and GM-CSF increased at 60 days p.i. and then decrease at 90 days p.i. [116]. On the other hand, CCL11 and G-CSF were increased at ten days p.i. and then returned to the baseline levels, whereas that IL-2 showed a biphasic increase (peak at three days p.i. and 60–90 days p.i.) and IL-10 was increased at 60 days p.i. in the striatum [116]. Finally, in the cortex, the levels of IL-1β, IL-9, CCL2, CCL11, CXCL1, CXCL10, and M-CSF have increased only the acute phase of the Influenza virus infection, whereas IL-2 and IL-10 presented a biphasic increase (peak at three days p.i. and 60–90 days p.i.) [116]. On the contrary, VEGF only was increased at 60–90 days p.i. [116]. The results obtained in this study suggest that the neurotropic strain H5N1 can induce a local innate immune response that involves the microglia. The neuroinflammation found in the acute phase of the influenza virus infection may be responsible for the encephalitis or encephalopathies observed in patients [116]. Moreover, the elevated levels of pro-inflammatory molecules after the infection underscore their role in developing neurodegenerative diseases [116]. Another neurotropic strain studied was H7N7 (rSC35M), which can cause long-term neuroinflammation. The data showed that infection with H7N7 promoted cognitive impairment at 30 days p.i. and caused an increase of IFN-γ and TNF-α in both sera and brains at eight days p.i. (Table 1) [113]. Moreover, H7N7 infection also altered the BBB permeability, promoting a strong local immune response that could be involved in the LTP impairment observed in the infected mice, suggesting that the pro-inflammatory molecules could play an important role in the normal function of these brain process [113]. Mouse cortical neurons were infected in culture with two strains of H1N1 (A/PR/8/34 and A/Shantou/169/2006) to evaluate the effects of influenza virus infection on CNS local cells, showing that both can infect neurons, but the replication was abortive [117]. However, the two strains of Influenza virus-induced a differential increase in the levels of IL-6, TNF-α, and CXCL-10 (Table 1) [117]. Furthermore, similar results were found in human neurons and astrocytes cell lines infected with H5N1 (A/Hong Kong/483/97), in which increased levels of IL-6 and TNF-α were found (Table 1) [118]. The avian H7N9 (H7/SH2/13) also can infect human neurons and astrocytes cell lines, and thus the infection results in the increase of IL-6, IL-8, TNF-α, IFN-β, and CCL2 (Table 1) [119]. These results suggest that neurons and astrocytes may contribute to the neuroinflammation caused by Influenza virus infection.

A study in a neonatal model evaluated whether influenza virus infection might influence brain function [120]. Mice were intraperitoneally infected with the mouse-adapted H1N1 (NWS/33) strain, detecting viral RNA in different brain zones, including the hippocampus, midbrain, the cerebellum, and the cerebral cortex, of neonatal mice as well in the CSF from adult mice [120]. Furthermore, increased mRNA levels of TNF-α, IL-1β, and IL-6 were found at five days p.i. simultaneously with the viral load peak. Interestingly, TNF-α mRNA levels decreased under the baseline at 15 days p.i. as compared to control mice, whereas IL-6 was decreased at 15 and 21 days p.i. as compared to control mice [120]. On the contrary, IL-1β mRNA levels were increased at 21 days p.i. (Table 1) [120]. These data suggest that the neuroinflammation directly caused by influenza virus infection may occur at different ages [120]. More information about the implication of neonatal infection and the possible neurological alteration in adulthood needs to be investigated.

Not only the neuroinflammation caused by influenza virus infection can be a result of neuroinvasion, but also by a systemic effect. H1N1 A/PR/8/34 is not a neurotropic virus but can induce a spatial learning impairment in the acute phase of infection, which may be caused by increased mRNA levels of pro-inflammatory cytokines, such as TNF-α, IL-1β, and IL-6 in the hippocampus, where the cognitive process takes place (Table 1) [114]. Additionally, the viral infection induces a decrease in the mRNA levels of neurotrophic factors, such as BDNF and NGF, which are essential to the neurons′ function and survival. The microglial activation may lead to the neuroinflammation observed that is consistent with the decreased mRNA levels CD200 and CX3CL1, which maintain the resting state of these cells and the increased microglia reactivity in the hippocampus of Influenza virus-infected mice [114]. H1N1 cannot cause long-term CNS alterations [113]. The non-neurotropic strain H3N2 (maHK68) also can cause an impairment of spatial learning and also alter the LTP [114]. This influenza virus strain has the ability to disrupt the BBB permeability and showed increased levels of TNF-α in the hippocampus at eight days p.i. [114]. The fact that non-neurotropic viruses can cause neurological alterations in animals can provide insights about potentially equivalent effects on patients.

According to the association between Influenza virus infection and SCZ disorder, several studies have evaluated how maternal infection with Influenza virus can be a risk factor for developing SCZ in the offspring [121,122,123]. A study suggested a higher risk of developing SCZ in the offspring when the mother was exposed to Influenza virus during the second trimester of pregnancy [124], but not with direct evidence [124]. Later, serological evidence associated the maternal exposure to Influenza virus during the first trimester of gestation with a higher risk of suffering SCZ by the offspring [125]. However, this association is still controversial in humans due to the study′s limitations [125]. According to the knowledge about the effects of the maternal immune activation (MIA), which induces systemic increased levels of pro-inflammatory molecules that may be triggered by the infection with influenza virus, can impair the brain development of the fetus [42,109]. However, the Influenza virus has not been detected in the brain of fetuses in an animal model, suggesting that the damage is an indirect effect of the infection [72,126]. H1N1 (A/WSN/33) infection during mid-pregnancy promotes an impairment in the locomotor activity in the offspring of mice and also affects the expression of 5-HT2A and mGlu2 in the frontal cortex, which are implicated in behavioral alteration [71,72]. Indeed, infection with this strain of the Influenza virus also promoted pyramidal neuron atrophy [71,72]. Additionally, maternal infection with H1N1 (A/NC-L/99) and H1N1 (A/WSN/33) promotes the loss of dopaminergic neurons in the offspring as well as behavioral alterations [110]. However, there is little evidence for an association between maternal infection with influenza virus, and the relationship with the development of SCZ, the influence of pro-inflammatory molecules was not addressed [110].

The potential association between maternal infection with influenza virus and an increased risk of developing ASD in the offspring remains unclear [102,127,128,129,130]. Abnormalities were found in the development of the brains of fetuses from influenza virus-infected mothers in animal models [71,123]. Similar findings were made for SCZ and influenza virus infection. There is no data about the pro-inflammatory molecule′s role triggered after maternal Influenza virus infection and ASD in the offspring.

### 2.2. Human Respiratory Syncytial Virus

Infants under two years old are the main targets of the human respiratory syncytial virus (hRSV) as well as the elderly and immunocompromised individuals, causing acute lower respiratory tract infection (ALTRI) [131,132,133]. The hRSV, recently renamed as human orthopneumovirus, is an enveloped, negative-sense singled stranded RNA virus [134]. The more severe clinical pathologies caused by the infection with hRSV are bronchiolitis and pneumonia [135]. hRSV infection leads to a strong inflammatory response in the airways, which is thought to establish a hyper-reactive immunological environment that fails to clear the virus [136]. Following virus entry into the respiratory tract, hRSV reaches alveoli and infects airway epithelial cells leading to the secretion of several pro-inflammatory molecules by epithelial cells, such as IL-1β, IL-6, CXCL8, IFNα/β, and TNF-α [136,137].

Moreover, the innate immune system’s activation can lead to massive infiltration of inflammatory cells into the lung parenchyma and alveoli [137]. The host response to the hRSV infection involves alveolar infiltration of neutrophils, eosinophils, NK cells, mDCs, plasmacytoid dendritic cells (pDCs), macrophages, B lymphocytes, CD4+ and CD8+ T cells [136,138]. Besides, lung pathology of hRSV infection is associated with a strong T helper 2 (Th2)-polarized T cell immunity against the virus [139] characterized by the production of cytokines such as IL-4, IL-5, and IL-13 [140]. These cytokines increase the recruitment of inflammatory cells, such as eosinophils, neutrophils, and monocytes, to the lungs and impair cytotoxic CD8+ T cells [141]. Several studies have suggested that hRSV could modulate the host adaptive immune response’s polarization to promote a detrimental Th2-biased T cell response. Moreover, the virus blocks the priming, expansion, and function of the Th1 and cytotoxic T cells needed for viral clearance [141,142,143]. Infection with this virus can also cause neurological alterations, such as seizures, encephalitis, and encephalopathy [144,145,146,147,148]. The detection of hRSV RNA in CSF has been reported in some cases, supporting the notion of neuroinvasion by this pathogen during the acute phase of the infection [147,149]. Besides, an elevated amount of IL-6 was found in CSF from patients with neurological signs and decreased levels of TNF-α, which correlates negatively with hRSV detection in CSF (Table 2) [150]. Furthermore, the detection of high levels of IL-8, CCL2, and CCL4 in CSF, also has been documented, which has no relation with the hRSV detection in CSF (Table 2) [150]. These findings suggest that not only hRSV can infect the CNS but also induce local inflammation directly or indirectly (Figure 2).

To better understand the neurological alterations caused by hRSV, studies in animals infected intranasally with strain 13018–8 of hRSV (a clinical isolate) showed that viral RNA and proteins were detected in several zones of the brain, such as the cortex and hippocampus, at three days p.i. [65]. The possible consequences of what may have caused the infection in the CNS were further evaluated by testing processes such as spatial learning and behavior 30 days p.i. The results showed impairment in both processes, demonstrating that the hRSV infection can have a long-term detrimental effect in these cognitive events [65]. Later, the ability of hRSV to infect CNS cells, including neurons was evaluated in vitro using the N2a cell line [151]. These cells not only were susceptible to the hRSV infection but also increased levels of IL-6 and TNF-α were detected in the supernatant of N2a hRSV-infected cells [151]. According to the fact that hRSV infection promotes learning and behavioral impairment and possibly infects neurons, new data showed that hRSV could disrupt the BBB, promoting immune cell infiltration [66]. Moreover, the behavioral alteration reporter previously in hRSV infected mice can be observed up to 60 days p.i. and also increased levels of anti-inflammatory molecules, such as IL-4 and IL-10, and pro-inflammatory molecules including CCL2 were found in the brain of these mice [66]. Moreover, GFAP was found elevated in hRSV-infected mice′s brains, suggesting that which suggest that are active and possibly causing chronic inflammation [66]. Besides infecting neurons, hRSV was able to infect endothelial cells, microglia, and astrocytes, being these last cells infected in a higher percentage [66]. Primary cultures of astrocytes infected with hRSV showed elevated amounts of anti-inflammatory molecules, such as IL-4, IL-10, and pro-inflammatory molecules including TNF-α during the first 12h p.i. whereas IL-6 was increased until 72h p.i. [66]. These results suggest that hRSV infection in the CNS promotes a local immune response leading to the production of pro-inflammatory molecules that may be important for the impairment of cognitive and behavioral processes. Besides, contrary to the knowledge about the pulmonary pathology caused by hRSV infection, the CNS alteration has been poorly studied.

As compared to other viruses such as influenza, no studies that can associate the hRSV with any neuropsychiatric disorder have been published yet. A recent study reported neurological effects even after the hRSV infection has been resolved [73]. This study evaluated in infants whether a severe episode of hRSV infection before six months of age could lead to long-term learning difficulties [73]. The acquisition of the native phoneme repertoire during the first year of age was evaluated. The data showed that infants who suffer a severe disease caused by hRSV infection presented poor distinction of native and nonnative phonetic contrasts at six months of age [73]. This poor distinction remained atypically sensitive to nonnative contrasts even after 12 months of infection, suggesting weak communicative abilities (Table 2 and Figure 2) [73]. This case is the first report about the long-term effect of hRSV infections in humans, showing possible neurological consequences of a viral respiratory infection. However, the mechanisms of cognitive impairment still need to be elucidated, as well as a potential association between the secretion of pro-inflammatory molecules and the cognitive and behavioral impairment observed in humans.

### 2.3. Severe Acute Respiratory Syndrome Coronavirus 2

The severe acute respiratory syndrome coronavirus 2 (SARS-CoV-2) is a new respiratory virus first identified in December 2019, during an outbreak of pneumonia in Wuhan, China [152,153]. SARS-CoV-2 was quickly disseminated to the entire planet, leading to a global pandemic that, up to date, remains uncontrolled. This virus belongs to the family *Coronavirus* and the genre *betacoronavirus* and has a non-segmented, single-stranded, and positive-sense viral RNA [154]. Since the identification of SARS-CoV-2, the virus has mutated and created several new variants (B.1.1.7., B.1.351, P.1 and B.1.526.) [155,156]. Up to the time this article was written, over 100 million positive cases and more than 2 million fatalities had been reported [157]. This virus severely affects the elderly and people with co-morbidities, such as hypertension, cardiovascular and respiratory diseases [158]. The most common symptoms are fever, myalgia, headache, and cough, and in severe cases, the patients can display respiratory distress and pneumonia [158]. Furthermore, the infection with SARS-CoV-2 has been associated with neurologic symptoms [158]. Several cases reported showing that patients with the infection displayed symptoms including encephalitis, anosmia, ageusia, and Guillain-Barré syndrome, among others [75,159,160]. Besides these cases, neuropsychiatric cases associated with the infection of SARS-CoV-2 have been reported, such as psychosis, delirium, neurocognitive syndrome, and mood disorders (Figure 3) [76,161].

Infection with SARS-CoV-2 can cause neurologic symptoms through direct contact—reaching the brain via olfactory nerves or hematogenous routes—or indirectly through the cytokine inflammatory storm [76]. Although none of these possibilities have been conclusively demonstrated, the genetic material of SARS-CoV-2 has been found in the CSF of patients with neurologic manifestation (demyelinating disease) [162]. The presence of the virus in the CSF suggests that SARS-CoV-2 can reach the CNS [162]. Post-mortem studies on patients infected with SARS-CoV-2 have revealed astrocyte activation, gliosis, microglial nodules, loss of myelin and axon degeneration, among other features [162,163,164,165]. Additionally, these studies revealed that various parts of the brain are affected due to the lung infection with SARS-CoV-2.

**Figure 3 pharmaceuticals-14-00340-f003:**
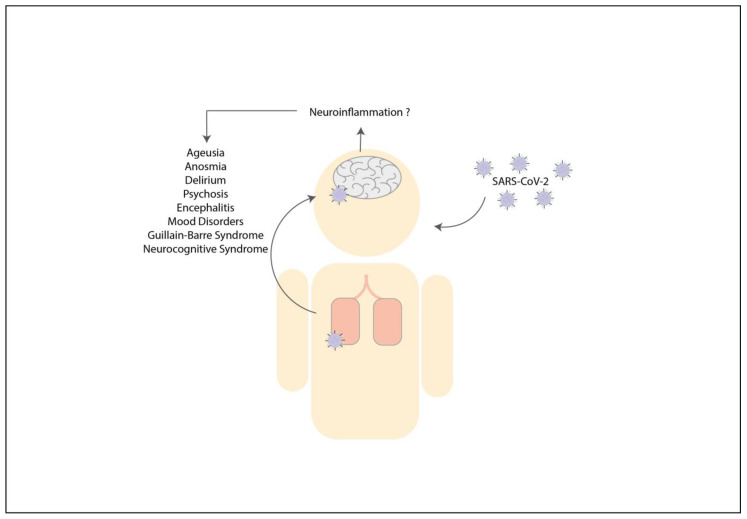
Infection with SARS-CoV-2 could cause neurologic consequences in patients. The respiratory infection with SARS-CoV-2 might lead to neuroinflammation as a consequence of a cytokine storm, which might increase the pro-inflammatory cytokines in the brain [74]. This increase might promote the development of neurologic disorders such as ageusia, anosmia, delirium, psychosis, encephalitis, mood disorders, Guillain-Barré Syndrome, and neurocognitive syndrome [75,76,159,164,165].

As mentioned above, patients infected with SARS-CoV-2 can develop neurological disorders through an inflammatory cytokine storm [74]. The cytokine storm is a term used when the immune system overacts due to the presence of an infection and leads to an increase in the secretion of inflammatory molecules systemically and to the bloodstream [166]. It has been reported that the infection with SARS-CoV-2 can cause a cytokine storm in patients with symptoms severe or critical, consisting of an increase of IL-1β, IL-4, IL-6, IL-10, IL-17, TNF-α, G-CSF, GM-CSF, IFN-γ, CCL2, CXCL8, and CXCL10 [167]. The increase in the inflammatory molecules has been related to cases of depression and anxiety in patients infected with SARS-CoV-2 [168]. In these cases, IL-6 was elevated when the symptoms appeared, and after the treatment -which decreased the inflammation- the neurologic symptoms were less notorious [168]. These cases suggest that the presence of specific inflammatory molecules can cause neurologic disorders. Unlike the previously described viruses, SARS-CoV-2 is a newly identified virus, and studies in animals analyzing this aspect have not been performed yet.

## 3. Potential Treatments to Decrease Neurologic Symptoms Caused by Infections

Patients infected by any of the previously mentioned viruses and show neurological symptoms are usually treated with antivirals and anti-inflammatory drugs (Figure 4). When an adult patient infected by influenza virus develops neurological symptoms, antiviral treatment has often been used [169,170,171]. Oseltamivir phosphate is an antiviral from the neuraminidase inhibitors drug used in treating patients infected with Influenza virus since this antiviral inhibits the reproduction of the virus [170]. During a study performed in 2009, 61.2% of the infants infected with influenza virus and with neurologic symptoms were treated with oseltamivir phosphate upon admission to the hospital, and because of this treatment, most of them have recovered without sequels [172]. Another antiviral drug that has been used is amantadine, which has the characteristic to penetrate the CSF [170]. In one case, infants infected with IAV received the antiviral treatment, and at the same time, were treated with methylprednisolone and ulinastatin -which are anti-cytokines drugs- and decreasing the temperature of the forehead [171]. Both patients were able to recover and did not present any sequels [171]. Other pro-inflammatory inhibitors that might be useful to reduce neuroinflammation are against TNF-α, IL-1β, and IL-6 [173]. It has been suggested that the use of therapies that suppress the pro-inflammatory molecules might be beneficial against SCZ as a complement of the regular treatment used in this disease [173]. The use of vaccines to protect against the complications developed due to the infection with influenza virus has been controversial. Some reports have indicated that certain vaccines against the influenza virus might be related to the development of neurologic disorders, while other reports have suggested possible protection against the development of symptoms like BGS [174,175]. If the vaccination against the Influenza virus protects from neurologic disorders, then the mother′s immunization during the pregnancy might prevent the neurologic symptoms in the offspring [176].

In the case of hRSV, there are no current treatments that have been tested in the population [67]. However, studies in animal models have suggested a possible way to prevent neurologic symptoms. In the study made in 2013, it was demonstrated that the vaccination of mice previous to the infection with hRSV could decrease the neurologic symptoms due to the infection with this virus [65].

For patients infected with SARS-CoV-2 that developed neurologic symptoms have been treated with antivirals. In the case of a patient positive to SARS-CoV-2 that presented acute myelitis, it was treated with several antivirals -such as ganciclovir and loponavir- along with antibiotics, such as moxifloxacin and meropenem [177]. Since this virus appeared over a year ago, not much has been advance in this area. However, probably suppressing the neuroinflammation might decrease the neurological symptoms. The evaluation as to how vaccines affect these symptoms remains unknown.

## 4. Conclusions

Pro-inflammatory molecules, such as cytokines and chemokines, play an essential role in the immune response and the immune and central nervous system′s proper function. IL-1β, IL-6, and TNF-α are the cytokines that have been constantly involved in the cognitive process. The pro-inflammatory molecules have been related to causing neuroinflammation, which might lead to neuropsychiatric disorders, including neurobehavioral alterations and cognitive impairment. Interestingly, viral infections have been thought to develop these alterations through direct infection of the CNS or increased local inflammation. Respiratory viruses are no exception to this; the influenza virus, hRSV, and SARS-CoV-2 have been associated with neuropsychiatric disorders. The Influenza virus infection can lead to neurologic symptoms by directly reaching the brain or increasing the brain′s inflammatory response. The pro-inflammatory molecules commonly increased during the infection with influenza virus are the IL-6, TNF-α, and IL-1β.

Additionally, the infection of pregnant women might result in neurologic alterations in the offspring due to the increase of pro-inflammatory cytokines in the mother. The hRSV infection can cause neurologic symptoms due to the virus′s possible neurotropic effect or the increase of inflammation in the brain. The pro-inflammatory molecules that increased in the CNS during the infection are most commonly IL-6 and CCL2. The infection with SARS-CoV-2 might cause neurologic symptoms since cases with neuropsychiatric symptoms have been related to an increase of pro-inflammatory cytokine IL-6. Whether this virus can reach the brain, or its effects are based on the pro-inflammatory response remains unclear. Notably, the symptoms developed due to these viral infections can be treated through the use of antivirals o preventing the pro-inflammatory response. It needs to be further evaluated the possibility of the prevention of these symptoms through the use of vaccines.

## Figures and Tables

**Figure 1 pharmaceuticals-14-00340-f001:**
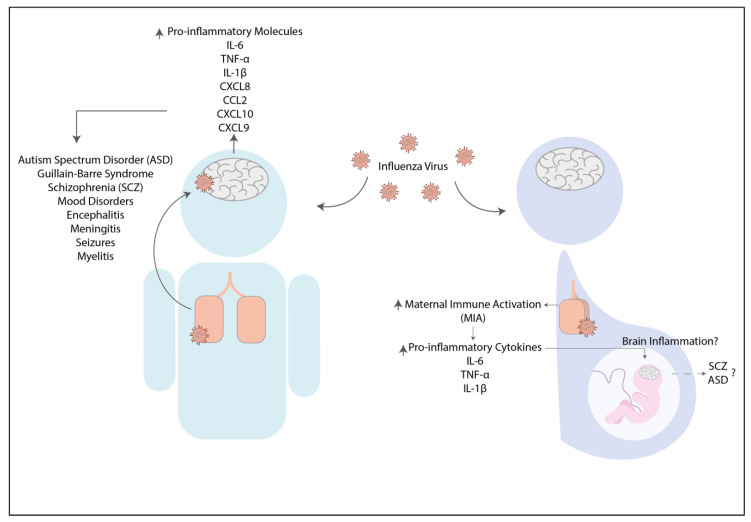
Infections with the influenza virus can cause neurologic consequences directly and indirectly. The respiratory infection with the Influenza virus can lead to neuroinflammation as a consequence of the elevated pro-inflammatory molecules, such as IL-6, TNF-α, IL-1β, CXCL8, CCL2, CXCL10 and CXCL9 [103,107,108]. An increase of these molecules can directly promote the development of neurologic symptoms such as schizophrenia (SCZ), autism spectrum disorder (ASD), myelitis, seizures, meningitis, encephalitis, mood disorders and Guillain-Barré syndrome. On the other hand, the respiratory infection with Influenza virus in pregnant women can increase maternal immune activation (MIA), which increases the secretion of pro-inflammatory cytokines, such as IL-6, TNF-α and IL-1β. An increase expression of these molecules might indirectly promote brain inflammation of the offspring in the uterus, impairing brain development [42,109]. A possible result of brain inflammation is SCZ and ASD [110].

**Figure 2 pharmaceuticals-14-00340-f002:**
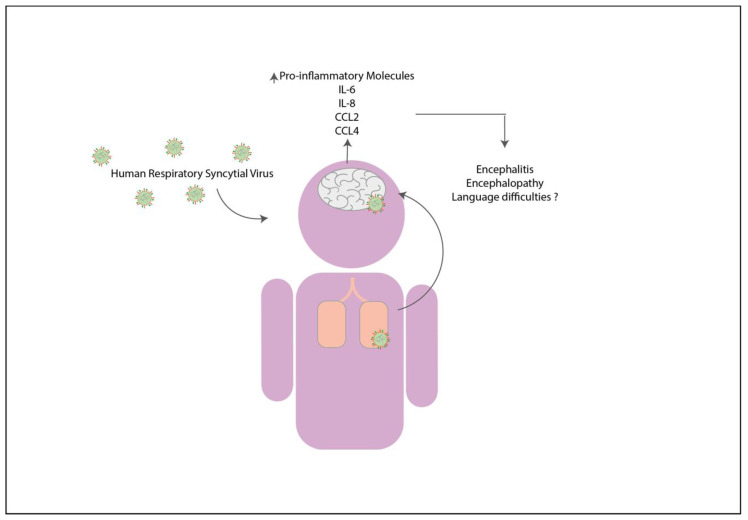
Human respiratory syncytial virus infection can cause neurologic consequences in patients. The respiratory infection with the respiratory syncytial virus can lead to neuroinflammation as a consequence of the elevated pro-inflammatory molecules such as IL-6, IL-8, CCL2, and CCL4 [150]. This increase can promote the development of neurologic symptoms such as encephalitis and encephalopathies [144,145,146].

**Figure 4 pharmaceuticals-14-00340-f004:**
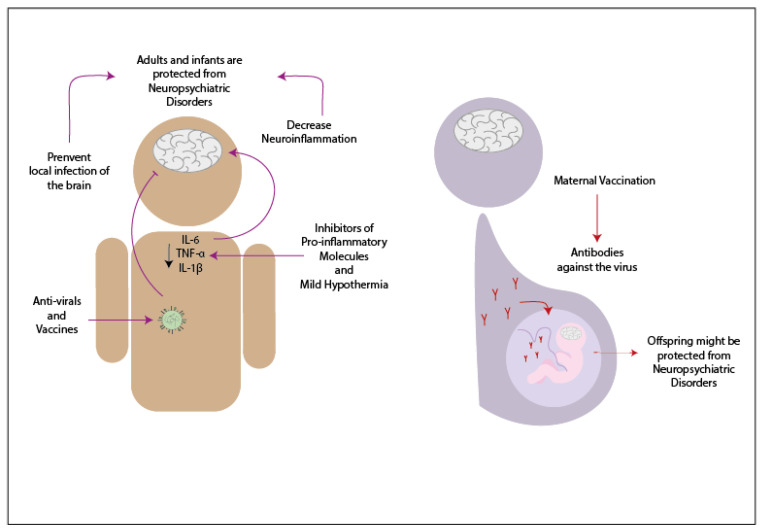
Possible treatments to prevent neurologic consequences in patients infected by respiratory viruses. Adults and infants that present diverse neuropsychiatric disorders can be decreased using antivirals drugs, which will prevent the brain′s local infection and therefore protect from further consequences. Additionally, the use of inhibitors of pro-inflammatory molecules and mild hypothermia can decrease the secretion of pro-inflammatory molecules, and as a result, decrease neuroinflammation. The use of vaccines might protect from developing neuropsychiatric disorders in adults, infants, and offspring due to the maternal vaccination.

**Table 1 pharmaceuticals-14-00340-t001:** Cytokine profile elicited by the influenza virus strains.

Strain	H5N1 (Vietnam/1203/04)	H7N7 (rSC35M)	H1N1 (A/PR/8/34 and A/Shantou/169/2006)	H5N1 (A/Hong Kong/483/97)	H7N9 (H7/SH2/13)	H1N1 (A/PR/8/34)	H3N2 (maHK68)
Neurotropic effect	Yes	Yes	Yes	Yes	Yes	No	No
Infection of cells	Neurons and microglia	No information	Neurons	Astrocytes	Neurons and astrocytes	No information	No information
Pro-inflammatory Cytokines	Increase of IL-1β, IL-12(p70), IL-2, IL-13, and TNF-α	Increase of IFN-γ and TNF-α	Increase of IL-6 and TNF-α	Increase of IL-6 and TNF-α	Increase of IL-6, IL-8, TNF-α and IFN-β	Increase of TNF-α, IL-1β, and IL-6	Increase of TNF-α
Chemokines	Increase of CCL2, CCL3, CCL4, CCL11, CXCL10, and CXCL1	No information	Increase of CXCL-10	No information	Increase of CCL2	No information	No information
Grown factors	Increase of G-CSF, GM-CSF, M-CSF and VEGF	No information	No information	No information	No information	Decrease of BDNF and NGF	No information
Anti-inflammatory Cytokines	Increase of IL-10	No information	No information	No information	No information	No information	No information

**Table 2 pharmaceuticals-14-00340-t002:** Cytokines elicited by hRSV infection in CNS and the associated consequences.

Clinical Findings	Pro-Inflammatory Molecules	Consequences
hRSV genetic material in cerebrospinal fluid	Elevated levels of IL-6, IL-8, CCL2, and CCL4 and low levels of TNF-α in cerebrospinal fluid	Encephalitis
		Encephalopathies
Learning difficulties	To be defined	Language difficulties

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
