# Peer review of "Contribution of Pro-Inflammatory Molecules Induced by Respiratory Virus Infections to Neurological Disorders"

_pharmaceuticals, 2021, doi:10.3390/ph14040340_

Round 1

Reviewer 1 Report

The authors have submitted a manuscript of illustrating a current overview regarding possible alterations of tissue contents of pro-inflammatory molecules such as cytokines and chemokines in the central nervous system (CNS) by virus infection such as influenza virus, human respiratory syncytial virus, and SARS-CoV-2. The authors claim that the alterations of the molecules are likely to induce neuroinflammation, which may lead to neuropsychiatric disorders. The authors searched a range of eligible literature, from well-known classical, neural, and molecular basis of pathology for neurodegenerative diseases to current knowledge of molecular pathways for releasing pro-inflammatory molecules in the CNS, resulting in reliable conclusions. This issue is of interest, especially under these unexceptional circumstances worldwide, and impact of their review is strong. My overall concern with the review describing the current available data regarding virus infection and the neuroinflammation induced by pro-inflammatory molecules is that information provided may offer something substantial that helps advance our understanding of novel medicines against CNS diseases induced by virus infection. The reference list may be useful for readers who are interested in this issue.

To better understand the possibility of pro-inflammatory molecule pathway usefulness, readers need to easily know the take-home messages from this review article: why and how these pathways could be considered to cure patients with the CNS diseases. For instance, a schematic diagram of putative molecular mechanisms of CNS diseases would strengthen this manuscript. The opposite, toxicological effects of these molecular pathways, if known, may influence largely the authors’ perspective.

Author Response

Answers to Reviewer 1:

  1. Reviewer 1: The authors have submitted a manuscript of illustrating a current overview regarding possible alterations of tissue contents of pro-inflammatory molecules such as cytokines and chemokines in the central nervous system (CNS) by virus infection such as influenza virus, human respiratory syncytial virus, and SARS-CoV-2. The authors claim that the alterations of the molecules are likely to induce neuroinflammation, which may lead to neuropsychiatric disorders. The authors searched a range of eligible literature, from well-known classical, neural, and molecular basis of pathology for neurodegenerative diseases to current knowledge of molecular pathways for releasing pro-inflammatory molecules in the CNS, resulting in reliable conclusions. This issue is of interest, especially under these unexceptional circumstances worldwide, and impact of their review is strong. My overall concern with the review describing the current available data regarding virus infection and the neuroinflammation induced by pro-inflammatory molecules is that information provided may offer something substantial that helps advance our understanding of novel medicines against CNS diseases induced by virus infection. The reference list may be useful for readers who are interested in this issue.

Answer: We would like to thank to the Reviewer for these comments.

  1. Reviewer 1: To better understand the possibility of pro-inflammatory molecule pathway usefulness, readers need to easily know the take-home messages from this review article: why and how these pathways could be considered to cure patients with the CNS diseases. For instance, a schematic diagram of putative molecular mechanisms of CNS diseases would strengthen this manuscript. The opposite, toxicological effects of these molecular pathways, if known, may influence largely the authors’ perspective.

Answer: As suggested by the Reviewer, we have added a new section explaining possible treatments to cure patients with CNS diseases and included a new figure with this information (Page 12, Lines 484-490).

We would like to thank the Reviewers and the Editor again for their time and effort to review this work. We hope that the current revised manuscript is acceptable for publication in Pharmaceuticals.

Reviewer 2 Report

Viral infection could be important causes of neuroinflammation and neurological disorders. Among the pathogenic mechanisms, inflammatory changes appear to have a pivotal role in diaese initiation and progression. In this manuscript, the involvement of respiratory virus infection in neuroninflammation and neurological disorders is highlighted.

  1. The content of current manuscript is helpful to the readers with interests in the interplay between respiratory virus infection and neuroninflammation, independent of viral CNS invasion. Regarding neuroinflammation, peripheral immune cells, BBB, and brain resident cells are of crucial roles in the initiation and propagation of inflammatory activation. Proinflammatory molecules are clearly described in current manuscript. However, the description of BBB and peripheral immune cells is limited. Relevant description and discussion of changes in peripheral immune cell population, polarization, and BBB barrier function are of importance to get a better understanding. Please add the relevant information.
  2. Discussion of any strategy for the prevention/treatment against proposed targets is interesting.

Author Response

Answers to Reviewer 2:

  1. Reviewer 2: Viral infection could be important causes of neuroinflammation and neurological disorders. Among the pathogenic mechanisms, inflammatory changes appear to have a pivotal role in diaese initiation and progression. In this manuscript, the involvement of respiratory virus infection in neuroninflammation and neurological disorders is highlighted.

The content of current manuscript is helpful to the readers with interests in the interplay between respiratory virus infection and neuroninflammation, independent of viral CNS invasion. Regarding neuroinflammation, peripheral immune cells, BBB, and brain resident cells are of crucial roles in the initiation and propagation of inflammatory activation. Proinflammatory molecules are clearly described in current manuscript. However, the description of BBB and peripheral immune cells is limited. Relevant description and discussion of changes in peripheral immune cell population, polarization, and BBB barrier function are of importance to get a better understanding. Please add the relevant information.

Answer: As suggested by the Reviewer, we have added more relevant information about BBB (Page 3, Lines 110-149).

  1. Reviewer 2: Discussion of any strategy for the prevention/treatment against proposed targets is interesting.

Answer: As suggested by the Reviewer, we have added a new section to discuss possible treatments (Page 12, Lines 458-502).

We would like to thank the Reviewers and the Editor again for their time and effort to review this work. We hope that the current revised manuscript is acceptable for publication in Pharmaceuticals

Reviewer 3 Report

7 March 2021

Review on the manuscript titled “Contribution of pro-inflammatory molecules induced by respiratory viruses infections to neurological disorders” by Bohmwald K et al, submitted to Pharmaceuticals

Dear Authors,

Behavioral abnormalities and cognitive impairments are caused by an immune response and associated with respiratory viral infections. The authors reviewed the proinflammatory response, behavioral abnormalities, and cognitive impairments triggered by infections of human respiratory syncytial virus, influenza, and SARS-COV-2. Please reconsider the following:

  1. Page 1, Abstract: Please present a rationale of choosing three viruses.
  2. Page 2, Lines61-68: Please update the references. Inflammatory responses in cognitive impairments were reviewed recently. Suggested references: Tanaka, M.; Bohár, Z.; Vécsei, L. Are Kynurenines Accomplices or Principal Villains in Dementia? Maintenance of Kynurenine Metabolism. Molecules202025, 564.  Tanaka, M.; Toldi, J.; Vécsei, L. Exploring the Etiological Links behind Neurodegenerative Diseases: Inflammatory Cytokines and Bioactive Kynurenines.  J. Mol. Sci. 202021, 2431.
  3. Pages 1-3, Introduction: The authors describe a wide range of illness. Please make it clear about acute and chronic inflammatory responses, pro-inflammatory and anti-inflammatory factors, and acute and chronic phase of clinical symptoms.
  4. Page 3, Lines 107-109: “… the most relevant …”; Please mention clearly why they are the most relevant.
  5. Page 4 Lines 146,147: “… inflammation of the offspring… “; Please refine the cation to describe that it happens in utero.
  6. Pages 7,8, Human respiratory syncytial virus: Please discuss neuropathic consequences by clearly separating the direct neural invasion, inflammatory factors, and their sequelae.

The manuscript contains three figures, one tables and 137 references. The reviewer recommends presenting more references, at least more than 150 for review articles and correcting the reference style. The figures and table are informative. Probably one more table can be added in the section of human respiratory syncytial virus. The list of abbreviation is missing. The manuscript carries important value regarding the immune response and neuropsychiatric symptoms under respiratory viral infection, but it deserves to clearly describe acute phase of infection, chronic phase of immune response, low-grade inflammation, adaptive immune response, and tolerogenic immune response, among others.  I reconsider this manuscript for publication after major revision.

Best regards,

Author Response

Answers to Reviewer 3:

  1. Reviewer 3: Behavioral abnormalities and cognitive impairments are caused by an immune response and associated with respiratory viral infections. The authors reviewed the proinflammatory response, behavioral abnormalities, and cognitive impairments triggered by infections of human respiratory syncytial virus, influenza, and SARS-COV-2. Please reconsider the following:

Page 1, Abstract: Please present a rationale of choosing three viruses

Answer: As suggested by the Reviewer, we had added the rationale of choosing these three viruses (Page 4, Lines 161-166).

  1. Reviewer 3: Page 2, Lines 61-68: Please update the references. Inflammatory responses in cognitive impairments were reviewed recently. Suggested references: Tanaka, M.; Bohár, Z.; Vécsei, L. Are Kynurenines Accomplices or Principal Villains in Dementia? Maintenance of Kynurenine Metabolism. Molecules 2020, 25, 564. Tanaka, M.; Toldi, J.; Vécsei, L. Exploring the Etiological Links behind Neurodegenerative Diseases: Inflammatory Cytokines and Bioactive Kynurenines. Mol. Sci. 2020, 21, 2431.

Answer: As suggested by the Reviewer, we had updated the references.

  1. Reviewer 3: Pages 1-3, Introduction: The authors describe a wide range of illness. Please make it clear about acute and chronic inflammatory responses, pro-inflammatory and anti-inflammatory factors, and acute and chronic phase of clinical symptoms.

Answer: As suggested by the Reviewer, we have clarified the description (Page 2, Lines 71,74-75,86-91,100; Page 3, Lines 101,102).

  1. Reviewer 3: Page 3, Lines 107-109: “… the most relevant …”; Please mention clearly why they are the most relevant.

Answer: As suggested by the Reviewer, we have mentioned clearly the relevance of each virus (Page 4, lines 162-167).

  1. Reviewer 3: Page 4 Lines 146,147: “… inflammation of the offspring… “; Please refine the cation to describe that it happens in utero.

Answer: As suggested by the Reviewer, we have modified the sentence (Page 5, Line 220).

  1. Reviewer 3: Pages 7,8, Human respiratory syncytial virus: Please discuss neuropathic consequences by clearly separating the direct neural invasion, inflammatory factors, and their sequelae.

 Answer: As suggested by the Reviewer, we have clarified the solicitated point (Page 8, Lines 373,374-375,376-377, 378; Page 9, Lines 387-388,398-401,404-405,408-413).

  1. Reviewer 3: The manuscript contains three figures; one tables and 137 references. The reviewer recommends presenting more references, at least more than 150 for review articles and correcting the reference style. The figures and table are informative. Probably one more table can be added in the section of human respiratory syncytial virus. The list of abbreviation is missing. The manuscript carries important value regarding the immune response and neuropsychiatric symptoms under respiratory viral infection, but it deserves to clearly describe acute phase of infection, chronic phase of immune response, low-grade inflammation, adaptive immune response, and tolerogenic immune response, among others.  I reconsider this manuscript for publication after major revision.

 Answer: As suggested by the Reviewer, we have added more references (181). Additionally, we have corrected the reference style. Also, we have added a table in the section of human respiratory syncytial virus (Table 2, Page10). According to the authors guidelines, do not require an abbreviation list. According to the available data, we have added more information about the immune response (Page 4, Lines 184-184; Page 8, Lines 353,369).

 We would like to thank the Reviewers and the Editor again for their time and effort to review this work. We hope that the current revised manuscript is acceptable for publication in Pharmaceuticals.

Round 2

Reviewer 1 Report

The authors have addressed properly all the issues raised by me.  I have no more comments, and now recommend that this manuscript is acceptable for publication in Pharmaceuticals.

Author Response

Reviewer 1: The authors have addressed properly all the issues raised by me.  I have no more comments, and now recommend that this manuscript is acceptable for publication in Pharmaceuticals.

Answer: We would like to thank the Reviewer for these comments and for their time and effort to review this work.

Reviewer 3 Report

27 March 2021

2nd review on the manuscript titled “Contribution of pro-inflammatory molecules induced by respiratory virus infections to neurological disorders” by Bohmwald K et al, submitted to Pharmaceuticals

Dear Authors,

Behavioral abnormalities and cognitive impairments are caused by an immune response and associated with respiratory viral infections. The authors reviewed the proinflammatory response, behavioral abnormalities, and cognitive impairments triggered by infections of human respiratory syncytial virus, influenza, and SARS-COV-2. Please reconsider the following:

  1. Page 1, Abstract: Please present a rationale of choosing three viruses. The rationale should be presented in Abstract.

The manuscript contains four figures, two tables and 181 references. The revised manuscript was substantially improved, but typos in the body of manuscript and references are expected to be corrected. The manuscript carries important value regarding the immune response and neuropsychiatric symptoms under respiratory viral infection. I recommend this manuscript for publication after minor revision.

Author Response

  1. Reviewer 3: Page 1, Abstract: Please present a rationale of choosing three viruses. The rationale should be presented in Abstract.

Answer: As suggested by the Reviewer, we had added the rationale of choosing these three viruses in the abstract (Page 1, Lines 21-25).

  1. Reviewer 3: The manuscript contains four figures, two tables and 181 references. The revised manuscript was substantially improved, but typos in the body of manuscript and references are expected to be corrected. The manuscript carries important value regarding the immune response and neuropsychiatric symptoms under respiratory viral infection. I recommend this manuscript for publication after minor revision

Answer: As suggested by the Reviewer, we had corrected the typos in the body of the manuscript and the references.

We would like to thank the Reviewers again for their time and effort to review this work. We hope that the current revised manuscript is acceptable for publication in Pharmaceuticals.